# Can situations awaken emotions? The compilation and evaluation of the Emotional Situation Sentence System (ESSS)

Yuan Zhao [1]☯*, Ming Yin[2]☯, Chuanlin Zhu[3], Chenghui Tan[4], Shengjie Hu[4], Dianzhi Liu[4]

1 Police Officer Academy, Shandong University of Political Science and Law, Jinan, China, 2 Jiangsu Police Institute, Nanjing, China, 3 School of Education, Soochow University, Suzhou, China, 4 School of Educational Science, Yangzhou University, Yangzhou, China

☯ These authors contributed equally to this work.
* psydzliu@163.com

**Data Availability Statement:** The data is available through the OSF at https://osf.io/8597f/, DOI: DOI 10.17605/OSF.IO/8597F.

## Abstract

We aimed to establish and evaluate a standardized emotional situation sentence system (ESSS) relevant to the lives of college students to supplement prior literature and adapt to the needs of emotional research. Two studies were designed for this research; study 1 examined the effect of words in the ESSS and study 2 involved the use of pictures. For Study 1, 778 items were selected by 607 college students and 15 experts. We then tested the scale with 80 undergraduate participants. The ESSS sentences were rated on their degree of valence, arousal, and dominance using a 9-point scale. Cronbach's α (greater than 0.986) of the overall score as well as each sub-score in the three components confirmed the scale's reliability. As seen on a scatter plot, the results suggest that negative emotions (fear, disgust, anger, sadness, anxiety) are convergent and different from the distribution of positive (happiness) and neutral emotions. Study 2 included 30 participants to compare the difference in valence and arousal between the ESSS and emotional pictures. The results indicate that the ESSS is a standardized, situational, and ecological emotional contextual text system, well-suited to invoke emotion in college students. The ESSS has significantly better arousal and potency than pictures; moreover, it can be applied to experimental studies of anxiety-related emotions. However, emotion pictures have shorter response times, and wider application ranges, and they can include more cross-cultural characteristics compared to words.

## Introduction

Everyone experiences a variety of emotions throughout their lives. Thus, the nature of emotion has always been of interest to scholars, as understanding the nature of emotion could lead to a clearer understanding of the human experience and yield useful results for improving people's emotional health [1]. However, the necessary standardization of stimulus materials presents a challenge in emotion research [2]. To address this issue, the National Institute of Mental Health (NIMH) research center for emotions and attention established a series of stimulating

**Funding:** This study was supported by the Qinglan Project of Jiangsu Universities and the police lie detection method with micro-expression recognition (Applied Innovation Project of Ministry of Public Security, Project No.: 2018YYCXJSST029).

**Competing interests:** No authors have competing interests.

material systems, including pictures, sounds, and words, that have been quantified and evaluated [2, 3]. As the study of emotion continues to grow, there are more complete picture-, sound-, image-, and word-emotion systems used in different countries. The "word-emotion system" is mainly the affective-words system, which is used primarily in the study of unconscious emotions [3, 4]. At present, many researchers are exploring the influence of positive and negative emotions on individuals' various cognitive activities by initiating explicit emotions [2–5]. Language is a potent tool for communicating emotions [6], but previous studies have found that words are less effective than pictures for arousing emotions [7, 8]. The key to effectively activating emotions is ensuring that participants become psychologically engaged in the corresponding situation because the situation constitutes a segment of life and has a good activation effect for exposing emotions [9]. Consideration of the need to psychologically engage emotions leads to the question, "What if we replace words with emotional statements?" Situational sentences may have a better priming effect on emotions than words, as browsing through situational sentences arouses more emotional responses [6].

Most depictions of the human emotion system in prior studies contain basic emotions, such as fear, disgust, anger, sadness, and happiness [10]; further, prior studies confirm that different emotions correspond to different qualities [1]. Anxiety is common among college students as they experience pressure from interpersonal relationships, life changes, environment changes, and academic and employment-related demands [11]. Moreover, college students are also easily affected by emergencies. This was confirmed by Hoyt et al. [12] in a survey of 707 American college students. The study results found that during the COVID-19 pandemic, most students experienced anxiety and stress, while their happiness continued to decline. Additionally, anxiety among college students is likely to lead to depression [13]. Therefore, it is crucial to understand anxiety through research. However, some studies have reported that pictures, sounds, and words cannot effectively activate an individual's anxiety, on their own [14]. This is possibly because anxiety is a compound emotion. For instance, in a picture, the context of the picture and the faces of individuals may not be easily identifiable in a snapshot alone. Additionally, while a scene in a movie may trigger anxiety, a single frame does not provide the full experience of the film. However, some studies indicate that words may be effective in triggering anxiety [15]. As situational statements can present varied and complex emotions with brief descriptions, in addition to the six basic emotions, we added "anxiety" to the present study to explore the activation of different emotion types more comprehensively. The first aim of the study was to examine the effectiveness of a new ESSS in eliciting emotional responses in college students. To verify the validity of ESSS, it is crucial to compare it with other emotional materials. However, few studies have achieved direct comparisons of emotional effects between stimulus domains [7]. Therefore, the second aim of this study was to compare the utility of the ESSS to that of emotion-inducing pictures and explore the advantages of the different materials.

## Design and methods

We designed two studies to explore the activation of emotions. The purpose of Study 1 was to explore a new mechanism of stimulating explicit emotions using emotional statements. To establish an ESSS relevant to college students, Study 1 drew upon standard procedures used by previous scholars to examine emotional systems [16]. At present, the emotion picture is one of the most commonly used emotion priming materials in emotion research, and contains a wide range of emotion types, with good reliability and validity [17, 18]. Study 2 aimed to examine the calibration validity of the ESSS as well as ecological validity. As emotional pictures and ESSS contain similar types of emotions, they have the possibility of comparison, and the

comparison with authoritative materials can better illustrate their respective advantages. The study was conducted by comparing the items with the highest arousal in the five basic emotions of the ESSS and the emotional pictures through behavioral experiments. The respective advantages and disadvantages of the ESSS and emotional pictures were analyzed in detail to provide directions for future research.

## Study 1

### Participants

The participants of Study 1 were divided into two groups. The first group, 607 undergraduates and postgraduates randomly selected from Jiangsu province (mainly Suzhou and Nanjing) completed the emotion-inducing situations questionnaire. These participants included 234 men and 373 women aged 21.57 ± 4.19 ($M \pm SD$) years. The second group graded the sorted materials. This group comprised 91 undergraduates randomly selected from a university in China, including 23 men and 68 women aged 18.19 ± 0.61 ($M \pm SD$) years. 11 participants with incomplete answers were excluded, and the number of final participants was 80, including 17 men and 63 women, aged 18.19 ± 0.62 ($M \pm SD$) years. According to the suggestion of the reviewer, 30 participants (10 men and 20 women) aged 20.07 ± 2.16 were randomly selected to reevaluate all the items to test the retest reliability. All participated in this study voluntarily and signed an informed consent form before we began the experiment. They were each paid 10 yuan after the experiment. All participants were healthy, without mental illness, and had normal or corrected-to-normal vision. This study was approved by the Ethics Committee of our school and conformed to ethical standards.

### The compilation process of the ESSS

**Collection of situation statements.**   A collection of situation statements was compiled from the first group's responses to the emotion-inducing situations questionnaire. This questionnaire consists of seven questions on seven dimensions of the situational emotion system (see S1 Text for details.). An example of a questionnaire item is, "Please describe, in one sentence, a situation that makes you feel sad, such as 'My beloved grandmother died suddenly' (please write down at least three such sentences)." In the first round, we filtered and modified the collected situational statements, deleting sentences that had already been documented. To make the meaning of the sentence clear to the participants quickly [19, 20], we ensured that each situational statement comprised approximately 10–15 words.

**Filtering situation statements.**   The collected situational statements were screened by 15 experts, including one professor, two associate professors, six doctoral students, and six master's students. Determining whether the items in each dimension evoked the appropriate emotions was the most important part of the selection process. If any one expert found an item to be inappropriate, all experts discussed and made a decision regarding deletion of the item. All the filtered statements involved common situations, had clear semantics, were not repetitious, and could stimulate the corresponding emotions in individuals. Finally, a total of 778 situational statements were selected, among which 121 reflected situations corresponding to fear, such as "waking up late at night looking at a dark room"; 119 corresponded to disgust, such as "walking into a room full of smelly garbage"; 111corresponded to anger, such as "someone read my diary without permission"; 94 corresponded to sadness, such as "my beloved grandparents passed away suddenly"; 116 corresponded to happiness, such as "I'm going to marry the one I love"; and 118 corresponded to anxiety, such as "I broke a valuable vase at my teacher's house by accident"; the remaining 99 statements were neutral, such as "I saw a few people studying in the classroom." The content mainly included scenes of campus life, family life,

family relationships, classmate relationships, loving relationships, nature scenes, and other scenes closely related to contemporary college students' lives.

**Rating situation statement.** According to Osgood's theory, we used a self-report method to evaluate the material from the three components of valence, arousal, and dominance [21, 22]. All participants were assessed for each emotional dimension. To reduce the fatigue effect of participants, the same participant completed the assessment of seven emotional dimensions in seven sessions, each lasting approximately 15 minutes. Participants completed the assessment through an electronic questionnaire. To understand the feelings of the participants more accurately, we asked them to rate the relationship of each item to the three components using a 9-point Likert-type scale (1 = very slightly/not at all; 9 = extremely). Specifically, in the valence component, a score of 1 indicates "extremely unpleasant" (very painful, annoyed, dissatisfied, sad, disappointed), and 9 indicates "extremely happy" (extremely happy, happy, satisfied, hopeful). In the arousal component, 1 means "extremely calm" (calm, relaxed, little stimulation, little attention), and 9 means "extremely uncalm" (extremely excited, exciting, interesting, alarming, bright). In the dominance component, 1 indicates "completely controlled" (completely affected by the content of the sentence and feeling weak, directed, controlled, manipulated), and 9 indicates "completely in control" (feeling dominant, having full control, fully restricted, influential).

## Results

**The reliability analysis.** The second group evaluated 778 sentences from three components: valence, arousal, and dominance. The Cronbach's α of the total average score of each of the three components were 0.987, 0.987, and 0.988, respectively, indicating a high level of credibility. We conducted reliability analysis based on previous studies [2, 19]. The Cronbach's α for the results of the seven emotional situational categories ranged from 0.986–0.998, and the split-half ranged from 0.895–0.986, indicating that the ESSS has high credibility. As shown in Table 1, the retest reliability was represented by the correlation coefficient between the two assessments, and the retest reliability of the ESSS is 0.981, 0.938 and 0.726.

**Descriptive statistical analysis.** Table 1 also reported a descriptive statistical analysis of 778 situational sentences in three components. Each sentence was rated by 80 participants. Then, the average score of 80 participants for a sentence was used as the sentence's score. The mean (*M*) focused on the average of the sentences, and the standard deviation (*SD*) focused on the differences between the sentences. For the total, *SD* indicated the consistency of the score; moreover, dominance was the smallest, indicating that the score was more uniform than valence and arousal. The range of valence and arousal was larger than dominance, which indicated that they have a wider range of scores.

In terms of different types of emotions, first, we analyzed negative emotions, among which there were four basic emotions. In terms of *SD*, the four emotions of fear, disgust, anger, and sadness showed little difference in the three components, that is, their scores were relatively consistent. For fear, the range of the three components was similar. For disgust, dominance was the most extensive and had the largest range. For anger, valence was the most extensive and had the largest range. In sadness, arousal was the narrowest and the range was the smallest. Anxiety is a complex emotion; it is rated low on valence, and can be considered a negative emotion. Among positive emotions—happiness had the highest *SD* of arousal. In terms of range, the arousal was the widest and the range was the largest. Finally, neutral sentences were analyzed, and the *SD* of valence was the largest, indicating that arousal and dominance degree were more concentrated than valence. In terms of range, the valence was the widest and the range was the largest.

**Table 1. Descriptive statistical analysis and retest reliability of the ESSS.**

| Type | Dimension | Range | First ($M \pm SD$) | Second ($M \pm SD$) | Retest reliability |
|---|---|---|---|---|---|
| | valence | 6.32 | 3.93 ± 1.88 | 3.80 ± 2.03 | 0.981** |
| Total | arousal | 4.17 | 6.22 ± 1.07 | 6.43 ± 1.29 | 0.938** |
| | dominance | 4.63 | 4.63 ± 1.04 | 4.71 ± 0.96 | 0.726** |
| | valence | 1.63 | 2.78 ± 0.32 | 2.73 ± 0.59 | 0.843** |
| Fear | arousal | 1.54 | 6.88 ± 0.26 | 7.12 ± 0.52 | 0.824** |
| | dominance | 1.47 | 3.89 ± 0.30 | 4.77 ± 0.69 | 0.777** |
| | valence | 1.58 | 2.68 ± 0.36 | 2.59 ± 0.55 | 0.856** |
| Disgust | arousal | 1.35 | 6.73 ± 0.33 | 6.65 ± 0.48 | 0.830** |
| | dominance | 1.96 | 4.02 ± 0.40 | 4.15 ± 0.75 | 0.832** |
| | valence | 1.00 | 2.59 ± 0.20 | 2.65 ± 0.51 | 0.766** |
| Anger | arousal | 0.89 | 6.95 ± 0.17 | 6.96 ± 0.39 | 0.655** |
| | dominance | 0.89 | 3.70 ± 0.19 | 4.32 ± 0.60 | 0.472** |
| | valence | 2.09 | 2.96 ± 0.41 | 2.57 ± 0.53 | 0.803** |
| Sadness | arousal | 1.74 | 6.47 ± 0.40 | 7.27 ± 0.51 | 0.776** |
| | dominance | 2.01 | 4.18 ± 0.45 | 4.00 ± 0.84 | 0.790** |
| | valence | 2.60 | 3.12 ± 0.55 | 2.58 ± 0.44 | 0.731** |
| Anxiety | arousal | 2.00 | 6.16 ± 0.43 | 6.61 ± 0.44 | 0.519** |
| | dominance | 2.29 | 4.68 ± 0.45 | 4.15 ± 0.47 | 0.603** |
| | valence | 1.30 | 7.58 ± 0.20 | 7.76 ± 0.36 | 0.717** |
| Happiness | arousal | 1.61 | 6.40 ± 0.35 | 6.77 ± 0.49 | 0.807** |
| | dominance | 1.23 | 5.33 ± 0.28 | 5.57 ± 0.63 | 0.680** |
| | valence | 1.62 | 5.92 ± 0.36 | 5.85 ± 0.47 | 0.739** |
| Neutral | arousal | 0.77 | 3.65 ± 0.16 | 3.34 ± 0.43 | 0.646** |
| | dominance | 0.52 | 6.83 ± 0.11 | 5.96 ± 0.28 | 0.113 |

1. $^*p < 0.05$,

$^{**}p < 0.01$,

$^{***}p < 0.001$.

2. "First" is the first assessment, which was scored by 80 participants. "Second" is the second assessment which was scored by 30 participants.

**Scatter plot analysis.** *Scatter plot analysis of total.* Ggplot 2 package in R language was used for scatter plot analysis [23, 24]. First, the scatterplots of total average scores of valence, arousal, and dominance were analyzed. As shown in Fig 1, the score distribution is relatively wide; however, there is a phenomenon of grouping, which may be due to different emotional dimensions (negative, positive, and neutral). Therefore, we will specifically analyze the score scatter diagrams for different types of emotion.

*Scatter plot analysis of negative emotions.* In the ESSS, fear, disgust, anger, sadness, and anxiety belong to the category of "negative emotions." As reported in Fig 2, the scatter diagram of negative emotions shows a similar distribution.

For fear, valence and arousal were significantly negatively correlated ($r$ = -0.84, $p < 0.001$) —the higher the valence, the lower the arousal. Arousal and dominance were significantly negatively correlated ($r$ = -0.79, $p < 0.001$)—the higher the dominance, the lower the arousal. Dominance and valence were significantly positively correlated ($r$ = 0.80, $p < 0.001$)—the higher the pleasure, the higher the dominance.

For disgust, the valence and arousal were significantly negatively correlated ($r$ = -0.87, $p < 0.001$)—the higher the valence, the lower the arousal. Arousal and dominance were

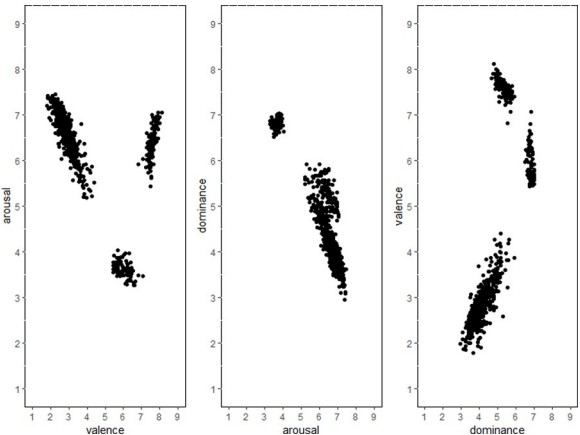

**Fig 1. Scatterplot of total.**

significantly negatively correlated ($r$ = -0.90, $p < 0.001$)—the higher the dominance, the lower the arousal. Dominance and valence were significantly positively correlated ($r$ = 0.86, $p < 0.001$)—the higher the valence, the higher the dominance.

For anger, valence and arousal were significantly negatively correlated ($r$ = -0.72, $p < 0.001$) —the higher the valence, the lower the arousal. Arousal and dominance were significantly negatively correlated ($r$ = -0.59, $p < 0.001$)—the higher the dominance, the lower the arousal. Dominance and valence were significantly positively correlated ($r$ = 0.52, $p < 0.001$)—the higher the valence, the higher the dominance.

For sadness, valence and arousal were significantly negatively correlated ($r$ = -0.93, $p < 0.001$)—the higher the valence, the lower the arousal. Also, the higher the dominance, the lower the arousal; thus, arousal and dominance were significantly negatively correlated ($r$ = -0.90, $p < 0.001$)—the higher the dominance, the lower the arousal. However, dominance and valence were significantly positively correlated ($r$ = -0.89, $p < 0.001$)—the higher the valence, the higher the dominance.

For anxiety, valence and arousal were significantly negatively correlated ($r$ = -0.89, $p < 0.001$)—the higher the valence, the lower the arousal. Also, the higher the dominance, the lower the arousal; thus, arousal and dominance were significantly negatively correlated ($r$ = -0.90, $p < 0.001$)—the higher the dominance, the lower the arousal. Furthermore, dominance and valence were significantly positively correlated ($r$ = 0.82, $p < 0.001$)—the higher the valence, the higher is the dominance.

*Scatter plot analysis of positive emotions.* As shown in Fig 3, for happiness, there was a significant positive correlation between valence and arousal ($r$ = 0.66, $p < 0.001$)—the higher the valence, the higher the arousal. Arousal and dominance were significantly negatively correlated ($r$ = -0.73, $p < 0.001$)—the higher the dominance, the lower the arousal. Dominance and valence were negatively correlated ($r$ = -0.67, $p < 0.001$)—the higher the valence, the lower the dominance.

*Scatter plot analysis of neutral emotions.* As shown in Fig 4, for neutral, valence and arousal were significantly negatively correlated ($r$ = -0.49, $p < 0.001$)—the higher the valence, the lower the arousal. Arousal and dominance were significantly positively correlated ($r$ = 0.25, $p < 0.05$)—the higher the dominance, the higher the arousal. Dominance and valence were negatively correlated ($r$ = -0.33, $p < 0.05$), the higher the valence, the lower the dominance.

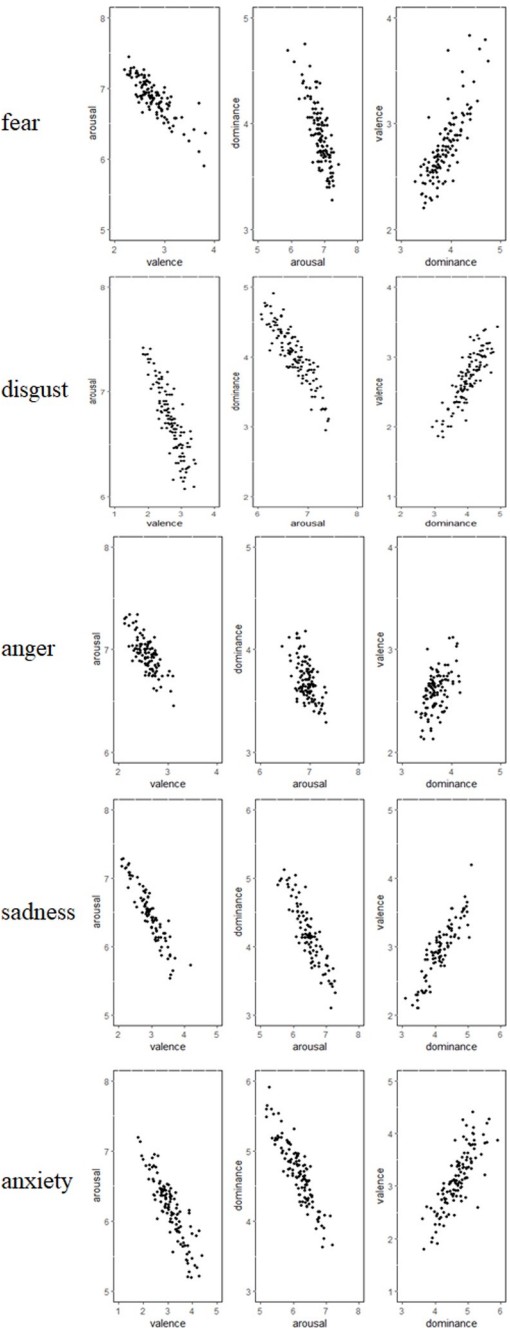

**Fig 2. Scatterplot of negative emotions.**

## Study 2

### Participants

Thirty participants (14 men and 16 women) from Suzhou, aged 20.10 ± 1.32 ($M \pm SD$) years were recruited through advertisements. The sample size was calculated by G*Power ($\alpha = 0.05$, $\beta = 0.8$), and the results showed that the sample size was standard and sufficient [25]. Participants volunteered for this experiment and had not participated in Study 1. They signed

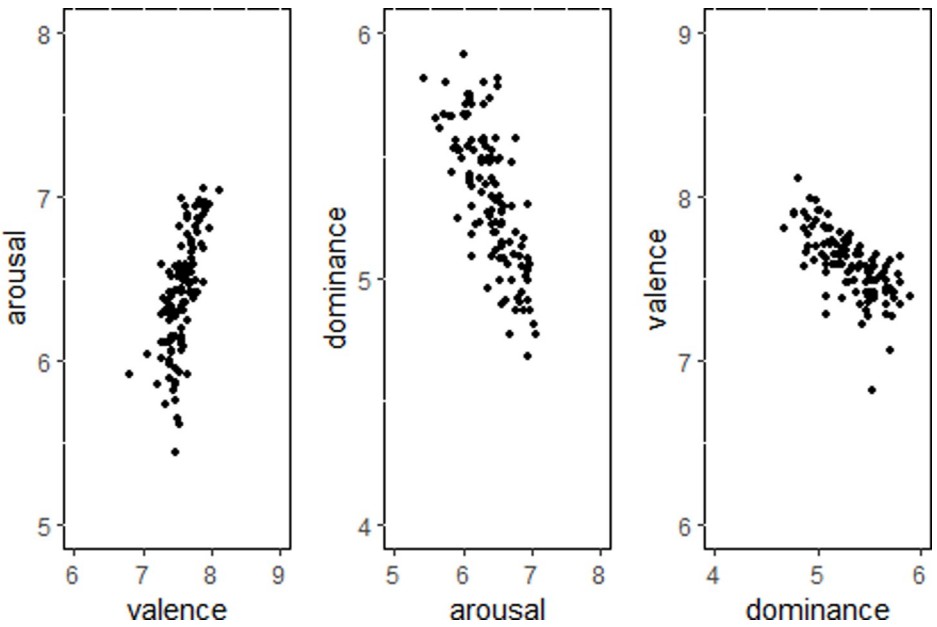

**Fig 3. Scatterplot of positive emotions.**

informed consent forms prior to the experiment and were paid 10 yuan as compensation on completing the experiment. In addition, 39 college students (25 men and 14 women) from Jiangsu Province, aged 20.54 ± 1.83 ($M \pm SD$) years completed the supplementary questionnaire. All participants were healthy, without mental illness, and had normal or corrected-to-normal vision. This study conformed to ethical standards and was approved by the Ethics Committee of our school.

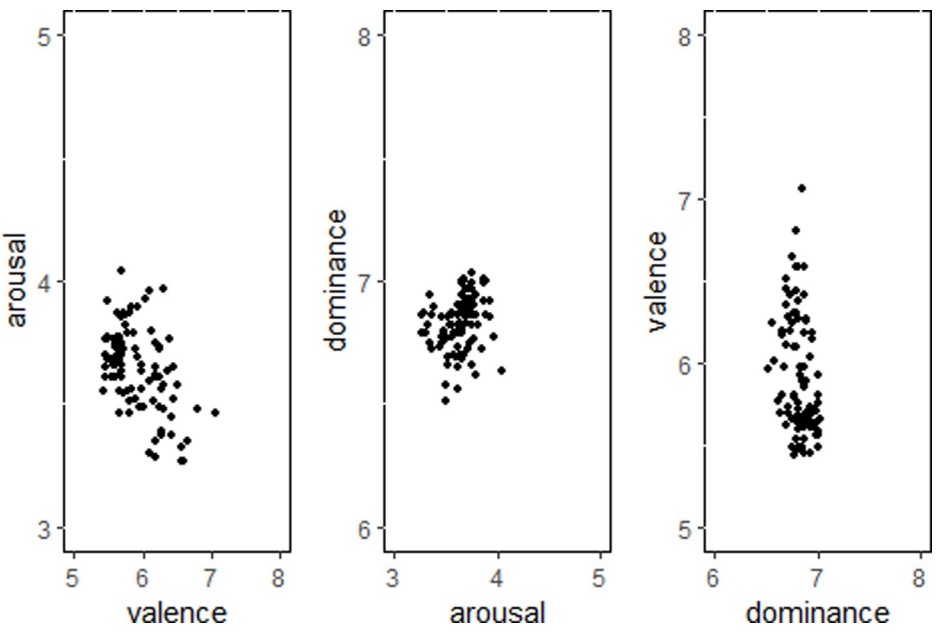

**Fig 4. Scatterplot of neutral emotion.**

## Experimental material

**Emotional pictures.** In this study, a total of 156 pictures were selected from the NimStim facial expression system [26]. Twenty-six pictures were selected for each of the six emotions: fear, disgust, anger, sadness, happiness and neutral; another four were randomly selected for the exercise stage. The model in this system had expressions with their mouth open as well as with their mouth closed. Langeslag, Gootjesand van Strien [27] found the former expression to be better as it could cause the individual to pay attention, and induce emotional effect. Therefore, the picture of the model with their mouth open was chosen in this study.

To ensure that the selected pictures could more effectively stimulate the corresponding emotions among participants, before the final experiment, 31 participants, aged 18.35 ± 0.93 ($M \pm SD$), were randomly selected from a university in Suzhou. They were asked to rate the pictures with the opened mouth in the NimStim facial expression system for arousal and valence. Finally, 13 pictures with the highest arousal for men and women in each dimension were selected as the formal materials of the experiment. The following details the selected picture number.

Each emotional dimension has its own code. The fear dimension included 13 pictures of women (No. 1, 3, 5, 7, 8, 9, 10, 11, 13, 14, 16, 18, 19) and 13 pictures of men (No. 21, 22, 26, 29, 30, 34, 35, 37, 38, 39, 40, 42, 43). The disgust dimension 13 pictures of women (No. 2, 3, 5, 6, 7, 8, 9, 10, 13, 14, 16, 17, 19) and 13 pictures of men (No. 23, 24, 26, 29, 30, 31, 32, 33, 36, 39, 40, 41, 43). The anger dimension included 13 pictures of women (No. 1, 3, 6, 7, 8, 9, 10, 12, 13, 14, 17, 18, 19) and 13 pictures of men (No. 20, 22, 23, 25, 26, 27, 29, 31, 33, 34, 36, 38, 39).The sadness dimension included 13 pictures of women (No. 1, 3, 5, 6, 7, 9, 10, 11, 12, 13, 14, 18, 19) and 13 pictures of men (No. 20, 23, 24, 25, 27, 30, 31, 32, 33, 39, 40, 41, 42).The happiness dimension included 13 pictures of women (No. 1, 2, 3, 5, 6, 7, 8, 9, 11, 13, 14, 18, 19) and 13 pictures of men (No. 22, 23, 25, 27, 30, 31, 33, 34, 35, 36, 39, 40, 41). The neutral dimension included 13 pictures of women (No. 1, 2, 3, 5, 6, 7, 8, 9, 10, 13, 14, 16, 18) and 13 pictures of men (No. 21, 23, 26, 28, 29, 30, 32, 33, 35, 37, 38, 40, 41).

The average valence and arousal of fear were 3.05 ± 0.46 and 6.69 ± 0.34 respectively. The average valence and arousal of disgust were 2.61 ± 0.46 and 7.04 ± 0.37 respectively. The average valence and arousal of sadness were 2.95 ± 0.37 and 6.28 ± 0.45 respectively. The average valence and arousal of happiness were 7.08 ± 0.28 and 6.44 ± 0.44 respectively. The average valence and arousal of neutral were 4.49 ± 0.35 and 4.31 ± 0.28 respectively. This indicates that the pictures used in the study can effectively enable the participants to experience the corresponding emotions and activate the participants' strong corresponding emotional.

**ESSS.** In ESSS, 26 emotional statements of fear, disgust, anger, sadness, happiness and neutral were selected respectively [28], and the other 4 were randomly selected for the exercise stage. The first 26 items with the highest arousal were selected from each emotional dimension [28]. The average valence and arousal of fear were 2.45 ± 0.14 and 7.20 ± 0.07 respectively. The average valence and arousal of disgust were 2.23 ± 0.23 and 7.10 ± 0.12 respectively. The average valence and arousal of anger were 2.44 ± 0.17 and 7.16 ± 0.10 respectively. The average valence and arousal of sadness were 2.51 ± 0.27 and 6.69 ± 0.19 respectively. The average valence and arousal of happiness were 7.78 ± 0.14 and 6.87 ± 0.10 respectively. The average valence and arousal of neutral were 5.77 ± 0.24 and 3.84 ± 0.08 respectively. Similarly, the statements used in the study can effectively make the participants experience corresponding emotions and activate their strong corresponding emotional experience.

## Procedure

The experiment was produced and presented using E-prime 2.0 (Psychology Software Tools, Inc., Sharpsburg, PA, USA). As shown in Fig 5, this experiment consisted of two types of tasks.

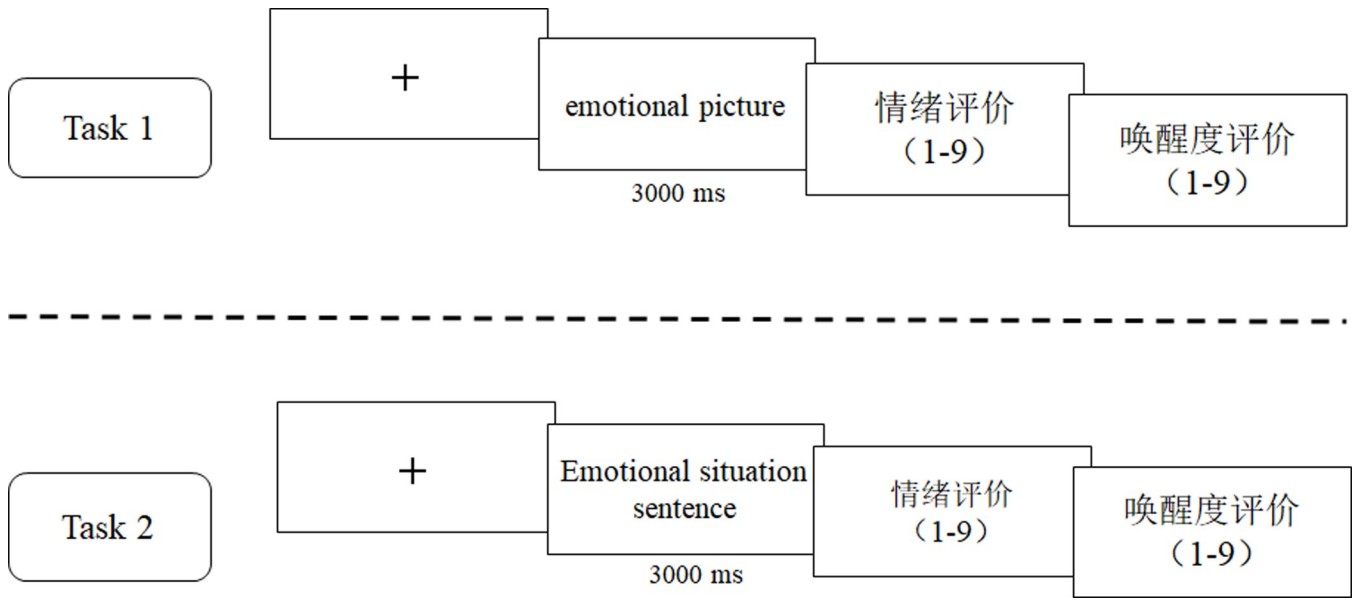

**Fig 5. The experimental procedure of study 2.**

In task one, a "+" lasting 500 ms was initially presented; next, the emotional pictures were presented. After 3000 ms, two emotional evaluation screens were presented. The first screen required participants to evaluate the emotional experience after looking at the picture. Ratings were given using a 9-point scale with "1" meaning extremely unpleasant (very painful, annoyed, dissatisfied, sad, disappointed), and "9" meaning extreme happiness (extremely happy, happy, satisfied, hopeful). The second screen required participants to evaluate arousal after looking at the picture. Arousal was also rated on a 9-point scale with "1" meaning extremely calm (calm, relaxed, little stimulation, least attention), and "9" meaning extremely uncalm (extremely excited, exciting, interesting, alarming, exciting, bright). Task two was similar to task one, except that the emotional statements from the ESSS were rendered in task two.

Before starting the experiment, the participants participated in practice experiments, and entered the formal experiments once they fully understood the experimental process. The participants were asked to complete six parts in total, each part belonging to an emotional dimension, including 26 pictures and 26 sentences, a total of 52 items. The six parts were presented randomly, and the 52 items in each section were also presented at random. At the end of the experiment, the participants were asked to answer two questions: 1. According to you, which picture is closer to college students' lives, the emotional picture, or the emotional situation statement?; 2. According to you, which picture is more interesting, the emotional picture or the emotional situation statement? The participants completed the experiment in approximately 30 minutes. To further compare the differences between emotional sentences and emotional pictures, 39 participants were recruited to complete the supplementary questionnaire after the experiment. In this questionnaire, the participants were shown two kinds of emotional priming materials, and then asked to evaluate how interesting the two materials were and how close they were to college students' lives. The higher the number, the more interesting/close they were.

## Results

**Descriptive statistical analysis.** In this study, the items with the highest arousal of five basic emotions in the ESSS and NimStim were compared. Table 2 reports the results of

**Table 2. Descriptive statistics of the different variables in seven emotions ($M \pm SD$).**

| type | variable | picture | sentence | t | Cohen's d |
|---|---|---|---|---|---|
| fear | valence | 3.72 ± 1.05 | 2.27 ± 0.62 | 6.57*** | 1.68 |
| | T-valence | 2107.37 ± 831.67 | 2337.44 ± 1144.38 | -1.17 | 0.23 |
| | arousal | 6.04 ± 1.33 | 7.74 ± 0.89 | -7.81*** | 0.85 |
| | T-arousal | 2024.64 ± 787.03 | 2124.05 ± 839.72 | -0.79 | 0.77 |
| disgust | valence | 3.32 ± 1.05 | 2.43 ± 0.73 | 5.13*** | 0.98 |
| | T-valence | 1279.78 ± 494.77 | 1804.09 ± 768.55 | -3.78** | 0.80 |
| | arousal | 6.33 ± 1.37 | 6.89 ± 1.04 | -2.85** | 0.46 |
| | T-arousal | 1184.28 ± 672.75 | 1785.55 ± 709.02 | -4.65*** | 0.87 |
| anger | valence | 2.85 ± 1.10 | 2.19 ± 0.86 | 3.16* | 0.67 |
| | T-valence | 965.30 ± 379.28 | 1398.83 ± 535.04 | -4.46*** | 0.93 |
| | arousal | 6.77 ± 1.57 | 6.38 ± 1.17 | -2.70* | 0.28 |
| | T-arousal | 915.07 ± 534.25 | 1267.20 ± 650.55 | -3.37** | 0.59 |
| sadness | valence | 3.89 ± 0.83 | 1.83 ± 0.82 | 12.28*** | 2.50 |
| | T-valence | 1016.37 ± 451.85 | 1453.39 ± 564.90 | -3.68** | 0.85 |
| | arousal | 5.63 ± 1.32 | 7.55 ± 1.31 | -10.39*** | 1.46 |
| | T-arousal | 862.49 ± 510.68 | 1322.56 ± 719.67 | -3.64* | 0.74 |
| happiness | valence | 6.06 ± 1.45 | 8.31 ± 0.99 | -7.29*** | 1.81 |
| | T-valence | 1181.79 ± 497.95 | 1437.30 ± 465.05 | -3.02** | 0.53 |
| | arousal | 5.27 ± 1.43 | 7.22 ± 1.40 | -8.86*** | 1.38 |
| | T-arousal | 1028.76 ± 570.22 | 1258.15 ± 636.73 | -2.26* | 0.38 |
| neutral | valence | 4.92 ± 0.59 | 5.37 ± 0.24 | -3.65*** | 1.00 |
| | T-valence | 841.85 ± 360.27 | 1247.17 ± 638.70 | -3.59** | 0.78 |
| | arousal | 4.42 ± 1.68 | 3.86 ± 1.56 | 2.15* | 0.35 |
| | T-arousal | 603.28 ± 418.64 | 905.98 ± 567.35 | -3.63** | 0.61 |

1. *$p < 0.05$,

**$p < 0.01$,

***$p < 0.001$.

2. "T" means time (ms), T-valence is the reaction time of valence, T-arousal is the reaction time of arousal.

emotional pictures and emotional sentences analysis using paired sample *t*-test. First, the participants' assessments of different negative emotions were analyzed. When different materials were used to stimulate the individual's fear, disgust, anger, and sadness, the valence of emotional sentences was significantly lower than that of emotional pictures. On the other hand, the arousal was significantly higher than that of emotional pictures. This result showed that when negative emotions were aroused, emotional sentences activate more unpleasant emotions in participants compared to emotional pictures. For reaction time, when the participants' negative emotions were activated, they took longer to judge the valence and arousal of sentences significantly.

Second, when different materials were used to stimulate participants' happiness, the valence and arousal of emotional sentences were significantly higher than that of emotional pictures. For the reaction times of valence and arousal, the reaction times for emotional sentences were significantly longer than those of emotional pictures.

Finally, when different materials were used to stimulate the participants' neutral emotion, the valence of emotional sentences was significantly higher than that of emotional pictures, but the arousal was significantly lower than that of emotional pictures. The reaction times for emotional sentences were significantly longer than those of emotional pictures, whether it was the judgment of valence or arousal.

**Analysis of essay questions.** Of the participants, 76.92% thought that emotional sentences were closer to college students' lives than emotional pictures, and 66.67% thought that emotional sentences were more interesting than emotional pictures. Paired sample $t$-test was used to analyze the differences between emotional sentences and emotional pictures in terms of being interesting and college life closeness. The results also indicated that emotional sentences (6.77 ± 1.39) were more interesting than emotional pictures (6.03 ± 1.93) ($t = 2.09$, $p < 0.05$, Cohen's $d = 0.44$), and emotional sentences (6.62 ± 1.84) were more relevant to college students' lives than emotional pictures (5.23 ± 1.95) ($t = 3.69$, $p < 0.01$, Cohen's $d = 0.73$).

## Discussion

### A standardized and ecological emotional situation sentence system is established

The results of the two studies indicate that the ESSS closely relates to college students' lives, and is suitable for emotional priming. The process of emotional material standardization includes collecting and selecting the material, determining the evaluation dimensions, implementing the evaluation, and analyzing the reliability of the evaluation and content. According to all the indexes of the measurement results, the ESSS conforms to the requirements for each measurement, with good reliability and validity. It is a standardized, situational, and ecological emotional situational sentence system that is different from the existing emotional priming materials.

In the process of standardization of the statements, according to Osgood's theory, the valence, arousal, and dominance of materials can be standardized by using the method of self-reported assessments [2, 19, 21, 22]. Through the Cronbach's α, the split-half and the retest reliability, we found that the reliability of each dimension was good and that the measurement requirements were met. Additionally, according to the scatterplot, all negative emotions showed a significant negative correlation between the valence and the arousal, a significant negative correlation between the arousal and the dominance, and a significant positive correlation between the dominance and the arousal. For positive emotions, there was a significant positive correlation between the valence and the arousal, a significant negative correlation between the arousal and the dominance, and a significant negative correlation between the dominance and the arousal. For neutral emotions, although there was significant correlation between the dimensions, the correlation coefficient obviously decreased. These results are similar to those reported by previous studies [19], indicating that the relationships are consistent with the rules of basic human emotions. The results also demonstrate that the seven emotions involved in this study could be further divided into three categories: negative emotions, positive emotions, and neutral emotions. This design could provide expedient conditions for scholars to explore positive and negative emotions, and neutral emotions can be used as a control group, increasing the scientific neutral of the research design.

The scientific nature of the ESSS is confirmed by the statistical indicators, was ensured by the selection principles for the situational statements, and is evident in their contents. First, regarding the selection principles, which is similar to previous studies [2, 19, 21], we found that, in principle, the ESSS requires items to express a clear meaning to quickly stimulate participants' corresponding emotions without requiring them to think carefully and consume too much of their cognitive resources. Second, the items in each emotion are drawn from the typical daily lives of the participants. Thus, they can easily arouse emotional resonance for the participants and stimulate the emotional experience. Third, we tried to enrich the content. For example, to elicit positive emotions, the ESSS included parents, friends, travel, study, money, work, games, exams, and so on. For negative emotions, the ESSS included things such as

buildings, animals, movies, food, friends, and parents. For eliciting neutral emotion, the ESSS included classrooms, books, the college campus, landscapes, sounds, plants, and so on. In the compilation of the emotional system, the richer the stimulus content, the better it is for inducing the emotions needed for the experiment [19]. At the same time, the rich scenes help maintain the interest of the participants, ensure the effect of emotional stimulation, and guarantee the effectiveness of emotional activation.

## The differences between ESSS and existing emotional activation materials

The ESSS differs from existing emotional priming materials in many ways. Previous studies also used emotional systems involving words, such as the affective norms for English words (ANEW) [21] and the Chinese emotion adjective words system [29]. However, while these emotional systems are related to words, they are often used to study unconscious emotions, especially implicit emotions [29]. Unlike a prior study, the present study established the ESSS using many statements with complete scenes. The purpose of the ESSS is the same as that of earlier systems using pictures, videos, sounds, and other emotional systems—to activate the explicit emotions of individuals. However, the ESSS is more likely to stimulate strong corresponding explicit emotions among participants in a short period due to the situational background. It is of great significance in examining emotion recognition of college students [30], the cultivation of college students' network social ability [31], and the neural mechanism of how emotion affects cognitive activities [32]. In addition, the items involved in this study are from college students' own lives as well as prior studies; thus, they are closely related to college students' lives, making the ESSS more ecologically relevant. Hence, it may be easier to activate the corresponding emotions of college students using the ESSS than by using other materials. It is undeniable that further revisions can be applied in the future, depending on the intended usage.

## The advantages of ESSS

Although the ESSS is an original emotional activation system, data analysis has demonstrated its many advantages over the emotional picture system in several ways. In valence, all negative emotions (fear, disgust, anger, and sadness) showed that the valence of emotional sentences was significantly lower than that of the pictures (the lower the valence, the less happy the participants felt); however, for happiness, the valence of emotional sentences was significantly higher than that of the pictures. From the perspective of arousal, the arousal of the six emotions compared in this study was significantly higher than that of the emotional pictures. With the emotional sentences, when the three emotions were activated, arousal was better than it was with pictures; moreover, no significant difference between the two was found in other dimensions. This finding suggests that emotional sentences are more effective, regardless of whether negative or positive emotions are needed. The results of negative emotions are different from the results of the comparison of words and pictures [7], which may be the advantage of emotional sentences and emotional words.

 The increased effectiveness of the ESSS may indicate that the included items are more closely related to the lives of college students than the items used by other systems; hence, college students may have more emotional responses to this system. During their college-going age, individuals have a stronger sense of identity, which enables them to experience stronger emotional resonance prompted by the faces of strangers or strange environments [33]. Furthermore, some participants may find the ESSS content more interesting than emotional pictures. In the survey, some participants reported that "emotional sentences are more attractive," "sentences make people more comfortable, and pictures are more rigid," and so forth. When

people are attracted to interesting objects, they tend to have higher levels of motivation [34], which may be one of the reasons why the ESSS can activate participants' emotions better than pictures.

Second, from the dimensions measured, the ESSS not only effectively activates the basic emotions described above but also can activate an individual's anxiety emotions. In fact, there are few studies on the emotion of anxiety evoked by ecological conditions and situations, even though tension is known to be a common negative emotion among college students [35]. At the beginning of college, the strange environment can bring out anxiety around interpersonal communication. College students must attain professional knowledge, and the diverse range of examinations they must pass may produce academic anxiety. Participating in campus activities such as speeches, debates, and music performance can create further anxiety relating to being on stage. Anxiety in college students must be properly controlled so that they can perform better in their studies, lives, and work, and they could enjoy the experiences of college lives.

Previous studies on anxiety in college students have often been conducted through questionnaires [36] and situational tests (such as Trier Social Stress Test, TSST) [37]. Due to the complexity and diversity of anxiety, there are few perfect methods for initiating anxiety in the subjects in experiments. The description of the development of the ESSS and its use may enrich the literature and make up for the gaps in previous studies, providing standardized tools for future scholars to explore anxiety, which is one of the innovations and the highlights of the ESSS.

Finally, from the researchers' point of view, it is necessary to evaluate the valence and arousal degree of the instrument, no matter which tool is used to activate the subjects' emotions [19]. When using previously established methods for eliciting emotion, men and women participants should be matched with gender-relevant emotional pictures [28]. However, using ESSS to activate emotions does not involve preparing different stimuli based on gender because there is no difference between men and women in the design of the situations. Thus, the linking of gender matching could be reduced, allowing the overall workloads of researchers to be reduced.

## The advantages of emotional pictures

According to the results of this study, although emotional sentences are better for activating emotions in valence and arousal, the use of emotional pictures still has some advantages. From the perspective of reaction time, when the participants evaluated valence and arousal, they responded significantly faster to the emotional pictures and the neutral pictures for the four emotions of disgust, anger, sadness, and happiness. This can be explained through the beliefs of some participants. For example, some participants believed that "pictures are more intuitive, sentences need more comprehension and understanding," and "emotional pictures can make my experience more direct." Prior studies on both emotions [38] and emotional regulation [39], that have explored experimental materials' priming effect on participants, and the differences between various emotional strategies on emotional regulation, usually require participants' subjective evaluations of their emotional experiences. The results of the present study show that the participants were able to judge their current emotional experience more quickly after seeing the emotional pictures. Moreover these quick responses could reduce the overall time of an experiment, especially during ERP experiments. Another important factor is that compared with emotional situational statements, emotional pictures usually only need 1000 ms to arouse a participant's emotions [17]. However, it is obviously difficult to finish reading the emotional situation statements within 1000 ms. The presentation time of the emotional

situation statements in this study was 4000 ms. Further verification is need to understand whether the emotion can be activated using emotional situation statements in a shorter time duration.

From the perspective of applicability, in previous studies, emotional pictures (such as the NimStim set of facial expressions used in this study as an example) have been widely applied to participants of all groups and ages. For instance, emotional pictures have been used to study patients with depression [17, 40], patients with schizophrenia [41], children [42], and college students [28]. Emotional pictures have a wide range of practicality, and when different groups see pictures of emotional faces, they can activate the corresponding emotions for all the groups. However, the projects involved in the ESSS are closely related to the lives of college students, so the ESSS is more suitable for research with college students as participants. Therefore, because its use is more targeted, its scope of application is inevitably narrowed.

From the perspective of cultural background, emotional pictures (also exemplified by the NimStim expression system) are widely used not only in Western studies [41, 42] but also in experimental research by Chinese scholars [40]. Emotional pictures show facial expressions, they can include racial diversity, and they can equally represent men and women [28]; therefore, their cultural background relevance is broad, which increases their applicability. However, the ESSS is more suitable for Chinese college students because only Chinese college students can recognize and understand some situations, such as those involving the CET-4 (a college English test used in China) and Taobao (an online shopping APP in China).

## Conclusion

The collection, screening, evaluation, and analysis of the ESSS are in line with the process of measurement. The ESSS comprises a set of standardized emotional scenario statements well-suited to the emotional activation of college students. The ESSS has significantly better arousal and potency than other stimuli and can be applied to experimental studies of "anxiety" emotions. However, using emotional pictures yields shorter response times, includes a wider application range, and may be better suited for examining cross-cultural characteristics. According to different research needs, researchers can choose the emotional initiation tool most suitable for their research.

This study also has some limitations. First, it follows the traditional research method where in the data is obtained from subjective participant reports. However, we can obtain more objective data if the scoring method is explained in greater detail and the physiological indexes of the participants are observed in future studies. Second, the ESSS is closely related to the lives of Chinese college students; therefore, there are certain limitations to its application. In the future, research tools suitable for middle school students, workers, college students in other cultures or countries, and other groups should be further developed to expand the research scope and the application population. In addition, although this study included seven emotions, the types of emotional priming, such as pride, surprise, and guilt, could be further expanded in future research. Finally, the ESSS is a newly established emotional statement library, and its effectiveness in activating emotions must be verified in more experimental studies. In the future, study samples can be expanded to provide more references for emotional problems of college students.

## Supporting information

**S1 Text. The emotion-inducing situations questionnaire.**
(DOCX)

## Acknowledgments

We wish to thank all participants for their participation and the reviewers for their valuable suggestions.

## Author Contributions

**Data curation:** Yuan Zhao, Ming Yin, Chenghui Tan, Shengjie Hu, Dianzhi Liu.

**Formal analysis:** Yuan Zhao, Shengjie Hu.

**Funding acquisition:** Ming Yin, Chuanlin Zhu, Dianzhi Liu.

**Investigation:** Yuan Zhao.

**Methodology:** Yuan Zhao, Dianzhi Liu.

**Software:** Yuan Zhao.

**Supervision:** Ming Yin, Chuanlin Zhu.

**Visualization:** Yuan Zhao.

**Writing – original draft:** Yuan Zhao.

**Writing – review & editing:** Yuan Zhao, Chuanlin Zhu, Dianzhi Liu.

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
