## [Decision Letter · Decision Letter 0]

12 Mar 2021

PONE-D-20-40437

Can situations awaken emotions? The compilation and evaluation of the Emotional Situation Sentence System (ESSS)

PLOS ONE

Dear Dr. Zhao,

Thank you for submitting your manuscript to PLOS ONE. After careful consideration, we feel that it has merit but does not fully meet PLOS ONE’s publication criteria as it currently stands. Therefore, we invite you to submit a revised version of the manuscript that addresses the points raised during the review process.

We look forward to receiving your revised manuscript.

Kind regards,

Zezhi Li, Ph.D., M.D.

Academic Editor

PLOS ONE

Journal Requirements:

2. Please change "female” or "male" to "woman” or "man" as appropriate, when used as a noun (see for instance https://apastyle.apa.org/style-grammar-guidelines/bias-free-language/gender).

4.Thank you for stating the following financial disclosure: [Yes].

"Yes"

6. We note that Figure 5 includes an image of a participant  in the study.

Reviewers' comments:

Reviewer's Responses to Questions

**Comments to the Author**

1. Is the manuscript technically sound, and do the data support the conclusions?

Reviewer #1: Partly

Reviewer #2: Yes

2. Has the statistical analysis been performed appropriately and rigorously? 

Reviewer #1: Yes

Reviewer #2: Yes

3. Have the authors made all data underlying the findings in their manuscript fully available?

Reviewer #1: No

Reviewer #2: Yes

4. Is the manuscript presented in an intelligible fashion and written in standard English?

Reviewer #1: Yes

Reviewer #2: Yes

5. Review Comments to the Author

Reviewer #1: This study established and evaluated a standardized emotional situation sentence system (ESSS) relevant to the lives of college students. This ESSS used many statements with complete scenes, which is innovative and necessary. The results showed that the ESSS could better stimulate the emotions of the subjects than emotional pictures.

I have some confusion and suggestions for modification.

(1)In study 1, there were two groups, and the first group put forward statements with complete scenes. However, I did not see any result of the first group in Results. A few examples for each emotion and expert-rating consistency reliability should present. The content in “Filtering situation statements” should be the results rather than methods. After all, the purpose of Study 1 is to build an emotional database, so the contents of this database should be the first and most important result. Then the results of reliability and validity of the database should be showed.

(2)I am confused by the Table 2. As I understand it, there are two kinds of analysis to get M and SD.

The first one is as follows: Each emotion had many sentences, just like a questionnaire dimension with a lot of questions. A participant rated all the sentences for this emotion, for example, fear. Then the average score of all sentences for fear is used as the participant's score for fear. Therefore, each participant had a score for fear. The M was obtained by taking the average of 80 participants, and the SD was obtained by taking the standard deviation of 80 participants. If you take this analysis approach, the M will focus on the average of the 80 participants, and the SD will focus on the individual differences of the 80 participants.

The second one is as follows: Each emotion had many sentences, just like a questionnaire dimension with a lot of questions. Each fear sentence was rated by 80 participants. Then the average score of 80 participants for a fear sentence is used as the fear sentence's score. Therefore, each fear sentence had a score. The M was obtained by taking the average of 121 fear sentences, and the SD was obtained by taking the standard deviation of 121 fear sentences. If you take this analysis approach, the M will focus on the average of the 121 fear sentences, and the SD will focus on the differences of the 121 fear sentences.

There are slight numerical differences but substantial qualitative differences between the two analysis approaches. In my opinion, the second approach is more suitable for the purpose of building a database, such as getting the average number of comments per sentence, as the author showed in study 2. Please indicate which method was used, and if the first method was used, I suggest using the second or both methods.

In addition, only 80 participants rated, which was small sample size for building and testing a database. less. To avoid fatigue effects, each participant did not evaluate all the sentences, so how was the data analyzed? Then the number of raters for each sentence was reduced further, which had a great impact on the reliability and validity of the database. Therefore, it is suggested to increase the number of participants to at lest 300.

(3)Study 2 compared the differences between emotional sentences and emotional pictures, and found that the valence of emotional sentences was closer to the emotional type, had higher arousal degree, higher authenticity and higher interesting. Why only use proportion, not 9 point scale, to evaluate authenticity and interesting? Authenticity and interesting should be two of the core values of the database and quantified data should be obtained using a 9-point scale. Study 2 actually examined the calibration validity of the ESSS (its similarity to the emotional pictures) and ecological validity (better than the emotional pictures). Study 1 only examined reliability, not validity. Therefore, Study 2 should point out that it actually examined validity. In this way, study 1 and study 2 will form a unified whole for the purpose of building ESSS. Of course, to test validity, study 2 need to increase the participants to about 100.

(4)Previous studies also used emotional systems involving words, such as the affective norms for English words (ANEW) (Fairfield et al., 2017) and the Chinese emotion adjective words system (Lei & Zhang, 2013). So why this study did not compare ESSS and emotion words? Such studies should be supplemented to reflect the advancement, necessity and validity of ESSS.

Since this study needs to further supplement data analysis and experiments, it is suggested that the editorial department give the author 3 months to modify.

Reviewer #2: The ESSS compiled in this study is suitable for experiments with college students as subjects. At present, quite a lot of researches on emotion are based on college students, so it is of great significance to compile an emotion system suitable for college students. Study 2 illustrated their respective advantages through the comparison of ESSS and emotional pictures, which increased the reliability of the research results. The following questions need to be considered by researchers.

1. Study 2 mainly compared the emotional pictures with ESSS, and the reasons for the comparison should be explained in more detail.

2. The pictures presented in the research results are not clear enough. Figure 5 shows a Chinese sentence that needs to be translated into the manuscript. It is suggested that researchers reformat the figure 1 to figure 4 to make them more beautiful and clear.

3. Study 1 mentioned that the number of words in the situational statements consisted of approximately 10–15 words, please explain the rationale for that.

4. The innovation of Study 1, compared to existing database, is that added “anxiety” in building ESSS. It is suggested that researchers should give adequate explanations of why it's necessary.

5. The results of Study 2 mainly indicated that the ESSS had significantly better arousal and potency than pictures, and the emotion images had shorter response times. Their difference in practical application, however, is less discussed. And so, to supplement and perfect that is suggested.

6. In the process of writing, it is necessary to increase the amount of reference literature in recent three years.

7. To further improve the quality of the language, researchers can invite English major teachers to help with sentence by sentence modification, especially in the discussion section.

6. PLOS authors have the option to publish the peer review history of their article (what does this mean?). If published, this will include your full peer review and any attached files.

Reviewer #1: **Yes: **Jianxin Zhang

Reviewer #2: No

---

## [Author Response · Author response to Decision Letter 0]

24 Apr 2021

Reviewer #1: This study established and evaluated a standardized emotional situation sentence system (ESSS) relevant to the lives of college students. This ESSS used many statements with complete scenes, which is innovative and necessary. The results showed that the ESSS could better stimulate the emotions of the subjects than emotional pictures.

I have some confusion and suggestions for modification.

(1)In study 1, there were two groups, and the first group put forward statements with complete scenes. However, I did not see any result of the first group in Results. A few examples for each emotion and expert-rating consistency reliability should present. The content in “Filtering situation statements” should be the results rather than methods. After all, the purpose of Study 1 is to build an emotional database, so the contents of this database should be the first and most important result. Then the results of reliability and validity of the database should be showed.

Reply: Thanks for the expert's suggestion. According to the experts' suggestion, we give an example of each emotion in filtering situation statements. This will allow readers to have a more direct understanding of the ESSS. Similar to previous studies (e.g. Bai et al., 2005; Liu et al., 2006; Fairfield et al., 2017), in this part, 15 experts are required to select and retain suitable items according to the principles.

(2)I am confused by the Table 2. As I understand it, there are two kinds of analysis to get M and SD.

The first one is as follows: Each emotion had many sentences, just like a questionnaire dimension with a lot of questions. A participant rated all the sentences for this emotion, for example, fear. Then the average score of all sentences for fear is used as the participant's score for fear. Therefore, each participant had a score for fear. The M was obtained by taking the average of 80 participants, and the SD was obtained by taking the standard deviation of 80 participants. If you take this analysis approach, the M will focus on the average of the 80 participants, and the SD will focus on the individual differences of the 80 participants.

The second one is as follows: Each emotion had many sentences, just like a questionnaire dimension with a lot of questions. Each fear sentence was rated by 80 participants. Then the average score of 80 participants for a fear sentence is used as the fear sentence's score. Therefore, each fear sentence had a score. The M was obtained by taking the average of 121 fear sentences, and the SD was obtained by taking the standard deviation of 121 fear sentences. If you take this analysis approach, the M will focus on the average of the 121 fear sentences, and the SD will focus on the differences of the 121 fear sentences.

There are slight numerical differences but substantial qualitative differences between the two analysis approaches. In my opinion, the second approach is more suitable for the purpose of building a database, such as getting the average number of comments per sentence, as the author showed in study 2. Please indicate which method was used, and if the first method was used, I suggest using the second or both methods.

In addition, only 80 participants rated, which was small sample size for building and testing a database. less. To avoid fatigue effects, each participant did not evaluate all the sentences, so how was the data analyzed? Then the number of raters for each sentence was reduced further, which had a great impact on the reliability and validity of the database. Therefore, it is suggested to increase the number of participants to at lest 300.

Reply: Thanks to the expert for the question. Based on your suggestions, we made the following corrections:

A. “The second approach is more suitable for the purpose of building a database”. As suggested by expert, the method for data analysis we actually used was the second approach. In order to make the readers understand more clearly, we have carried on the improvement to this part as follows:

Table 2 reported a descriptive statistical analysis of 778 situational sentences in three components. Each sentence was rated by 80 participants. Then, the average score of 80 participants for a sentence was used as the sentence's score. The mean (M) focused on the average of the sentences, and the standard deviation (SD) focused on the differences between the sentences.

B. Maybe we didn't express it clearly enough before, which caused some misunderstanding. To reduce the fatigue effect of participants, the same participant completed the assessment of seven emotional dimensions in seven sessions, and each session last about 15 minutes. In other words, every participant rated all seven emotional dimensions and they only evaluated one emotional dimension every time which lasting about 15 minutes, instead of “each participant did not evaluate all the sentences”. The final number of participants was determined based on previous research on the compilation of emotional system (Bai, Ma, Huang, & Luo, 2005). Initially, 91 participants were enrolled in the study, but to improve the accuracy of the results, we eliminated participants who did not complete the seven assessments. The following is the number of participants participating in the evaluation in some of the studies. It can be found that the sample of our study is larger or similar to that of previous related studies.

1. Bai, L., Ma, H., Huang, Y. X., & Luo, Y. J. (2005). The development of native Chinese affective picture system-A pretest in 46 college students. Chinese Mental Health Journal, 19(22), 719-722.

Sample: 46; Cited: 400 times; Downloads: 7178

2. Liu, T. S., Luo, Y. J., Ma, H., & Huang, Y. X. (2006). The establishment and assessment of a native affective sound system. Psychological Science, 29(2), 406-408.

Sample: 50; Cited: 102 times; Downloads: 1587

3. Mikels, J. A., Fredrickson, B. L., Larkin, G. R., Lindberg, C. M., Maglio, S. J., & Reuter-Lorenz, P. A. (2005). Emotional category data on images from the international affective picture system. Behavior Research Methods, 37(4), 626-630.

Sample: 60; Cited: 320 times

4. Lei, Y., Sun, X. Y., Dou, H. R. (2019). Specifically inducing fear and disgust emotions by using separate stimuli: The development of fear and disgust picture systems. Journal of Psychological Science,42(03):521-528

Sample: 84; Cited: 1 times; Downloads: 645

(3)Study 2 compared the differences between emotional sentences and emotional pictures, and found that the valence of emotional sentences was closer to the emotional type, had higher arousal degree, higher authenticity and higher interesting. Why only use proportion, not 9 point scale, to evaluate authenticity and interesting? Authenticity and interesting should be two of the core values of the database and quantified data should be obtained using a 9-point scale. Study 2 actually examined the calibration validity of the ESSS (its similarity to the emotional pictures) and ecological validity (better than the emotional pictures). Study 1 only examined reliability, not validity. Therefore, Study 2 should point out that it actually examined validity. In this way, study 1 and study 2 will form a unified whole for the purpose of building ESSS. Of course, to test validity, study 2 need to increase the participants to about 100.

Reply: We carefully considered the expert's suggestion and improved the manuscript according to the expert's suggestion. 

A: The scale description is indeed not more accurate than the 9-point scale, so we re-surveyed 39 college students and asked them to compare the interestingness and familiarity of ESSS with emotional pictures using a 9-point scale, and the results are as follows.

Paired sample t-test was used to analyze the differences between emotional sentences and emotional pictures in terms of being interesting and college life closeness. The results also indicated that emotional sentences (6.77 ± 1.39) were more interesting than emotional pictures (6.03 ± 1.93) (t= 2.09, p < 0.05, Cohen's d= 0.44), and emotional sentences (6.62 ± 1.84) were more relevant to college students' lives than emotional pictures (5.23 ± 1.95) (t= 3.69, p < 0.01, Cohen's d = 0.73).

B: Study 2 actually examined the calibration validity of the ESSS (its similarity to the emotional pictures) and ecological validity (better than the emotional pictures). This suggestion from the expert is very accurate and useful. Therefore, we have made supplements and improvements in Design and Methods.

C. In terms of sample size, calculated by G* POWER 3.1 (α=0.05, β=0.8), the sample size of this experiment has reached saturation state (30 >27).In addition, our sample size was similar to, or even larger than the sample size of previous related studies. For example, through 17 participants (30>17), Xie and Yang (2016) compared the differences in emotional stimulation of four commonly used emotional-inducing methods: pictures, music, movies and meetings. Bayer and Schacht (2014) compared the difference between emotional words, pictures and faces with 24 participants in the experiment (30>24). Schacht, Adler, Chen, Guo and Sommer (2012) compared the difference between emotional pictures, faces and words with 16 participants (30>16).

1. Xie, Y. Z., Y, Z. (2016). A comparative study on the validity of different mood induction procedures (MIPs). Studies of Psychology and Behavior, 14(5), 591-599. (Sample: 17)

2. Bayer, M., & Schacht, A. (2014). Event-related brain responses to emotional words, pictures, and faces - a cross-domain comparison. Frontiers in Psychology, 5, 1106. (Sample: 24)

3. Schacht, A., Adler, N., Chen, P., Guo, T., & Sommer, W. (2012). Association with positive outcome induces early effects in event-related brain potentials. Biological Psychology, 89(1), 130-136. (Sample: 16)

4. Bayer, M., & Schacht, A. (2014). Event-related brain responses to emotional words, pictures, and faces - a cross-domain comparison. Frontiers in Psychology, 5, 1106. (Sample: 25)

5. Kensinger, E. A., & Schacter, D. L. (2006). Processing emotional pictures and words: Effects of valence and arousal. Cognitive, Affective, & Behavioral Neuroscience, 6(2), 110-126. (Sample: 21)

(4)Previous studies also used emotional systems involving words, such as the affective norms for English words (ANEW) (Fairfield et al., 2017) and the Chinese emotion adjective words system (Lei & Zhang, 2013). So why this study did not compare ESSS and emotion words? Such studies should be supplemented to reflect the advancement, necessity and validity of ESSS.

Reply: Thanks for the expert's suggestion. Previous studies used emotional systems involving words, such as the affective norms for English words (ANEW) (Fairfield et al., 2017) and the Chinese emotion adjective words system (Lei & Zhang, 2013). Although these emotional systems are related to words, they are often used to study unconscious emotions, especially implicit emotions (Lei & Zhang, 2013). ESSS plays a similar role to emotional pictures, and its main purpose is to activate the explicit emotions of the participants in the experiment. In particular, the complement of anxiety makes up for the deficiency of the previous emotional system. According to the suggestion of the expert, we made a supplement to “The differences between ESSS and existing emotional activation materials” on the basis of previous studies (Smith & Smith, 2019; Lin, Li, Cao, Lv, & Ke, 2018; Ding et al., 2020).

In the future research, we will pay attention to this problem. According to the suggestion of the expert, we explain the shortcomings of the study at the end of the manuscript.

Since this study needs to further supplement data analysis and experiments, it is suggested that the editorial department give the author 3 months to modify

Reviewer #2: The ESSS compiled in this study is suitable for experiments with college students as subjects. At present, quite a lot of researches on emotion are based on college students, so it is of great significance to compile an emotion system suitable for college students. Study 2 illustrated their respective advantages through the comparison of ESSS and emotional pictures, which increased the reliability of the research results. The following questions need to be considered by researchers.

1. Study 2 mainly compared the emotional pictures with ESSS, and the reasons for the comparison should be explained in more detail.

Reply: Thank you for your valuable suggestion. Based on previous studies (Cui, Song, Si, Wu, & Feng, 2021; Mowle, Edens, Ruchensky, & Penson, 2019), we clarified why the ESSS were compared with the emotional pictures, and the specific reasons are as follows.

At present, emotion picture is one of the most commonly used emotion priming materials in emotion research, and contains a wide range of emotion types, with good reliability and validity. Study 2 aimed to examine the calibration validity of the ESSS as well as ecological validity. As emotional pictures and ESSS contain similar types of emotions, they have the possibility of comparison, and the comparison with authoritative materials can better illustrate their respective advantages.

2. The pictures presented in the research results are not clear enough. Figure 5 shows a Chinese sentence that needs to be translated into the manuscript. It is suggested that researchers reformat the figure 1 to figure 4 to make them more beautiful and clear.

Reply: The expert's suggestion is very useful, the previous figures are really not clear enough. To better present our results, ggplot 2 (Wickham, 2010) package in R language (R Core Team 2020) was used for scatter plot analysis. 

Translating the Chinese in figure 5 will help more readers to understand the experimental process. However, as the editor proposed that the picture in Figure 5 should be modified, in order to correspond with the description in Task 1, we modified Figure 5, so this sentence in Chinese was deleted.

3. Study 1 mentioned that the number of words in the situational statements consisted of approximately 10–15 words, please explain the rationale for that.

Reply: According to the expert's suggestion, we have made a supplement to this part, as follows:

To make the meaning of the sentence clear to the participants quickly[19, 20], we ensured that each situational statement comprised approximately 10–15 words.

4. The innovation of Study 1, compared to existing database, is that added “anxiety” in building ESSS. It is suggested that researchers should give adequate explanations of why it's necessary.

Reply: Thanks for the expert's valuable suggestion. Anxiety is an important part of our study, we have made a supplement to this part, as follows:

Moreover, college students are also easily affected by emergencies. This was confirmed by Hoyt et al. [12] in a survey of 707 American college students. The study results found that during the COVID-19 pandemic, most students experienced anxiety and stress, while their happiness continued to decline.

5. The results of Study 2 mainly indicated that the ESSS had significantly better arousal and potency than pictures, and the emotion images had shorter response times. Their difference in practical application, however, is less discussed. And so, to supplement and perfect that is suggested.

Reply: The advantages of emotional pictures in reaction time are discussed in the first paragraph of the advantages of emotional pictures. According to the expert's suggestion, we have made a supplement to this part. We further demonstrate that priming participants' emotions with emotional pictures is more suitable for the study of neural mechanisms and can shorten the duration of the experiment.

6. In the process of writing, it is necessary to increase the amount of reference literature in recent three years.

Reply: In the process of this revision, we paid more attention to the suggestion of the expert when referring to previous studies, and revised the manuscript according to the research of the recent three years (e.g. Cui, Song, Si, Wu, & Feng, 2021; Ding et al., 2020; Hoyt, Cohen, Dull, Maker Castro, & Yazdani, 2021; Mowle, Edens, Ruchensky, & Penson, 2019).

7. To further improve the quality of the language, researchers can invite English major teachers to help with sentence by sentence modification, especially in the discussion section.

Reply: Thanks for the expert's suggestion. In order to improve the quality of the article language, we first conducted self-proofreading. Important parts of the manuscript were then revised through the Edtage, a professional language organization. The supporting materials are as follows.

 

The editor also put forward a lot of valuable suggestions, according to the editor's suggestion, we have revised the manuscript.

Reply: We revised the manuscript at the editor's request. As this is my first submission to PLoS ONE, I am worried that there is something wrong with it. If there is any problem, I am very sorry. So, if the editor finds any problem, please feel free to contact me. I will try my best to modify it.

2. Please change "female” or "male" to "woman” or "man" as appropriate, when used as a noun (see for instance https://apastyle.apa.org/style-grammar-guidelines/bias-free-language/gender).

Reply: We have changed "female” or "male" to “woman” or "man" as appropriate.

3. PLOS requires an ORCID ID for the corresponding author in Editorial Manager on papers submitted after December 6th, 2016.

Reply: According to the editor's requirement, I registered an account with ORCID.

4. Thank you for stating the following financial disclosure: [Yes].

a. Please clarify the sources of funding (financial or material support) for your study. List the grants or organizations that supported your study, including funding received from your institution.

d. If you did not receive any funding for this study, please state: “The authors received no specific funding for this work.”

Reply: This study was supported by the Qinglan Project of Jiangsu Universities and the police lie detection method with micro-expression recognition (Applied Innovation Project of Ministry of Public Security, Project No. : 2018YYCXJSST029).

5. We note that you have indicated that data from this study are available upon request.

Reply: The data of this study has been uploaded in the system. As a final note, the full version of ESSS is available via the link. You can also email the corresponding author for ESSS and related data.

Reply: We deleted the people photo in Figure 5.

---

## [Decision Letter · Decision Letter 1]

20 May 2021

Can situations awaken emotions? The compilation and evaluation of the Emotional Situation Sentence System (ESSS)

PONE-D-20-40437R1

Dear Dr. Zhao,

We’re pleased to inform you that your manuscript has been judged scientifically suitable for publication and will be formally accepted for publication once it meets all outstanding technical requirements.

Kind regards,

Zezhi Li, Ph.D., M.D.

Academic Editor

PLOS ONE

Additional Editor Comments (optional):

Reviewers' comments:

Reviewer's Responses to Questions

**Comments to the Author**

1. If the authors have adequately addressed your comments raised in a previous round of review and you feel that this manuscript is now acceptable for publication, you may indicate that here to bypass the “Comments to the Author” section, enter your conflict of interest statement in the “Confidential to Editor” section, and submit your "Accept" recommendation.

Reviewer #2: All comments have been addressed

2. Is the manuscript technically sound, and do the data support the conclusions?

Reviewer #2: Yes

3. Has the statistical analysis been performed appropriately and rigorously? 

Reviewer #2: Yes

4. Have the authors made all data underlying the findings in their manuscript fully available?

Reviewer #2: Yes

5. Is the manuscript presented in an intelligible fashion and written in standard English?

Reviewer #2: Yes

6. Review Comments to the Author

Reviewer #2: (No Response)

7. PLOS authors have the option to publish the peer review history of their article (what does this mean?). If published, this will include your full peer review and any attached files.

Reviewer #2: No

---

## [Editor Report · Acceptance letter]

8 Jul 2021

PONE-D-20-40437R1 

Can situations awaken emotions? The compilation and evaluation of the Emotional Situation Sentence System (ESSS) 

Dear Dr. Zhao:

I'm pleased to inform you that your manuscript has been deemed suitable for publication in PLOS ONE. Congratulations! Your manuscript is now with our production department. 

Kind regards, 

on behalf of

Dr. Zezhi Li 

Academic Editor

PLOS ONE